# Prediction of Vacant Parking Spaces in Multiple Parking Lots: A DWT-ConvGRU-BRC Model

Liangpeng Gao [1,2], Wenli Fan [1], Zhiyuan Hu [1] and Wenliang Jian [1,3,*]

[1] School of Transportation, Fujian University of Technology, Fuzhou 350108, China
[2] School of Transportation, Southeast University, Nanjing 210036, China
[3] College of Transportation Engineering, Tongji University, Shanghai 201804, China
* Correspondence: wenliang_jian@fjut.edu.cn

**Abstract:** For cities, the problem of "difficult parking and chaotic parking" increases carbon emissions and reduces quality of life. Accurately and efficiently predicting the availability of vacant parking spaces (VPSs) can help motorists reduce the time spent looking for a parking space and reduce greenhouse gas pollution. This paper proposes a deep learning model called DWT-ConvGRU-BRC to predict the future availability of VPSs in multiple parking lots. The model first uses a discrete wavelet transform (DWT) to denoise the historical parking data and then extracts the temporal correlation of the parking lots themselves and the spatial correlation between different parking lots using a convolutional gated recurrent unit network (ConvGRU) while using a BN-ReLU-Conv (1 × 1) module to further improve the propagation and reuse of features in the prediction process. In addition, the model uses availability, temperature, humidity, wind speed, weekdays, and weekends as inputs to improve the accuracy of the forecasts. The model performance is evaluated through a case study of 11 parking lots in Santa Monica. The DWT-ConvGRU-BRC model outperforms the LSTM and GRU baseline methods, with an average testing MAPE of 2.12% when predicting multiple parking lot occupancies over the subsequent 60 min.

**Keywords:** parking prediction; deep learning; discrete wavelet transform; convolutional gated recurrent unit network; multiple parking lots





## 1. Introduction

With economic and population growth, motor vehicle ownership is growing rapidly, which has exacerbated the imbalance between the supply and demand of vacant parking spaces (VPSs) in cities. Drivers typically spend 3.5–14 min looking for a VPS and cruising to find a VPS accounts for 8–74% of traffic [1]. Excessive time spent by drivers looking for VPSs increases time costs, fuel consumption, and emissions and leads to traffic congestion [2]. Parking difficulties and disorderly parking problems are often affected by accessibility, parking prices, and the number of VPSs.

To address this problem, some parking management and inducement systems have been developed to provide real-time VPS information [3–6]. These systems typically collect real-time available parking data using cameras and sensors [7,8]. In addition, several crowd-sensing-based schemes monitor the availability of street parking using mobile communication devices and in-vehicle sensors [9,10]. However, these parking guidance systems cannot guarantee the real-time nature of a VPS. That is, when a driver arrives at the designated parking space, the parking space may already be occupied. Due to the high cost of sensor equipment and maintaining real-time parking information, Y et al. [11] and Baidu Maps proposed a model named Du-parking to estimate the real-time parking availability of the whole city. Therefore, to enable car owners to purposefully find parking spaces, it is necessary to develop a parking guidance information system with predictive algorithms that can help drivers plan driving routes to find VPSs and reduce driving costs [12], which



can assist traffic planning and management to reduce energy consumption and traffic congestion [2].

In this paper, we propose a DWT-ConvGRU-BRC model. This model consists of a discrete wavelet transform (DWT) [13], convolutional GRU networks (ConvGRUs) [14], a two-layer linear network, and a composite function of three consecutive operations, i.e., batch normalization (BN), rectified linear activation (ReLU), and a $1 \times 1$ convolution (Conv), denoted BRC. First, we use the DWT to denoise the VPS data. Noise reduction before forecasting can eliminate the volatility of the VPS data themselves. Then, we use a deep learning-based prediction model that leverages ConvGRUs and a two-layer linear network to incorporate the spatial–temporal features of multiple data sources acquired in networks. Finally, the propagation and reuse of features in the prediction process are further improved using the BRC composite function.

This paper contributes to the literature in the following ways:

- We propose a deep learning-based parking space prediction model from the perspective of multiple parking lots. The model considers the processing of parking noise data as well as the spatial correlation of multiple parking lots and the temporal correlation of the parking lots themselves and uses a variety of factors, including parking lot occupancy, temperature, humidity, wind speed, weekdays, and holidays, to predict the number of available VPSs.
- Our proposed DWT-ConvGRU-BRC model can simultaneously predict the number of available parking spaces in multiple parking lots. Specifically, a ConvGRU is used to capture the spatial–temporal features of multiple parking lots, a two-layer linear layer is used to extract external influences, and BRC is used to further improve the propagation and reuse of features in the prediction process.
- The performance of the method is evaluated with a case study in the Santa Monica area. According to the results, the model outperforms other baseline methods, including LSTM, GRU, ConvGRU, and dConvLSTM-DCN models. Moreover, the results prove the improvement in prediction accuracy from the DWT and the effectiveness of incorporating weekday, vacation, and weather features into parking lot occupancy predictions.

The rest of this paper is organized as follows: Section 2 summarizes the literature review. Section 3 describes the detailed DWT-ConvGRU-BRC prediction model. Section 4 presents the results and analysis of the comparison experiments. Finally, we provide our conclusions and discuss possible future work in Section 5.

## 2. Literature Review

Access to VPS data has become easier with breakthroughs in sensor technology. However, the VPS data obtained in practical applications are often subject to different degrees of noise pollution. How to effectively process the data collected by sensors and improve the accuracy of algorithms is a thorny problem that many existing prediction methods still face. To solve this problem, wavelet analysis has been applied in some recent studies and has proven to be effective. For example, Li et al. [15] used the wavelet function for multiscale wavelet decomposition and reconstruction of VPS data using the hidden layer function of a wavelet neural network to improve prediction accuracy. Ji et al. [16] proposed a multistep prediction study of impacted parking spaces based on a WT in combination with a multistep prediction strategy using threshold noise reduction to further improve the prediction accuracy. Therefore, effective noise removal helps to improve the efficiency and accuracy of prediction.

Predicting the occupancy rate of multiple parking lots is one of the necessary links to solve the "difficult parking" problem. In recent years, VPS prediction has been divided into two categories: one is based on a statistical prediction model, and the other is based on machine learning (ML) and deep learning (DL). For statistical prediction models, Caliskan et al. [17] combined continuous Markov and queuing theory models to predict the occupancy status of parking lots in the destination area. On this basis, Xiao et al. [18]

proposed a continuous-time Markov M\M\C\C model for predicting available parking spaces. Caicedo et al. [19] proposed a real-time available dynamic algorithm using historical information to predict the availability of each parking lot. In addition, Rajabioun et al. [20] developed a vector spatiotemporal autoregressive model that can be used to predict the availability of parking spaces at a driver's estimated arrival time at both on-street and off-street parking locations. Peng et al. [21] modelled the discrete occupancy rate of a parking lot as a nonstationary Poisson process and proposed a cost-effective method for searching for parking spaces. Abdeen et al. [22] proposed a smart parking algorithm that varied the weights of five factors (availability, gate wait time, parking cost, traffic congestion, and driving distance to the parking lot) to achieve balanced traffic allocation and parking best use of the field. In fact, these statistical prediction models are highly dependent on assumptions about the arrival and departure process and therefore have difficulty adapting to the dramatic fluctuations in parking traffic flow.

For ML/DP prediction models, researchers have applied models such as regression trees, support vector machines (SVMs), support vector regression (SVR), neural networks, K-nearest neighbour (KNN), and random forests models to predict parking availability [23–27]. Hu et al. [28] combined support vector regression (SVR) and the fruit fly optimization algorithm (FOA) to predict the number of vacant parking spaces. Fan et al. [29] optimized a multi-step long short-term memory recurrent neural network (LSTM-NN) model with a grid search method to predict the number of vacant parking spaces. Moreover, many scholars have combined nonlinear system theory and optimization algorithms with neural networks to improve prediction accuracy. For example, Vlahogianni et al. [30] used a genetic algorithm-optimized multilayer perception (MLP) to predict the occupancy rate of a regional parking lot over the subsequent 30 min. Camero et al. [31] used a genetic algorithm (GA) combined with a recurrent neural network (RNN) to predict parking occupancy in Birmingham. Zeng et al. [32] combined a wavelet transform (WT) with bi-directional LSTM (Bi-LSTM) to further improve the prediction accuracy using threshold noise reduction.

In addition, some scholars have considered the influence of external factors, such as weather and holidays, on VPS forecasting. Fokker et al. [33] explored the influence of external factors such as weather on parking occupancy and found that external factors improved the predictive performance by 8%. In Zhang's [34] work, a PewLSTM was proposed for predicting parking behaviour by combining the effects of weather and parking periodicity. Zeng et al. [35] proposed a stacked gated recurrent unit (GRU)-LSTM model that combined the efficiency of a GRU and the accuracy of LSTM and incorporated various factors as inputs, such as weather, to predict the availability of parking spaces. ML/DL methods can automatically learn from past samples to better describe complex nonlinear problems. However, the ML/DP methods described above only consider the temporal correlation of VPS data and fail to consider the spatial correlation of VPSs in multiple parking lots.

Therefore, this paper proposes a model called DWT-ConvGRU-BRC to predict the number of VPSs in multiple parking lots. Our model combines the advantages of wavelet transform. While capturing the spatial–temporal correlation of multiple parking lot data, it also takes external factors such as weather as input to improve the accuracy of the model. Previous research related to our methodology includes DWT-Bi-LSTM [32] and dConvLSTM-DCN [36].

## 3. Methodology

### 3.1. Data Description

To evaluate the performance of the proposed prediction model, we conducted a case study in Santa Monica, CA, USA (longitude range: $[-118.499378, -188.49361]$, latitude range: $[34.019575, 34.010806]$) [37], which has 11 parking lots scattered over the road network, as illustrated in Figure 1. The data were collected from 6 April 2021 to 13 May 2021. The number of VPSs was collected every 5 min, resulting in 10,944 pieces of historical data per parking lot.

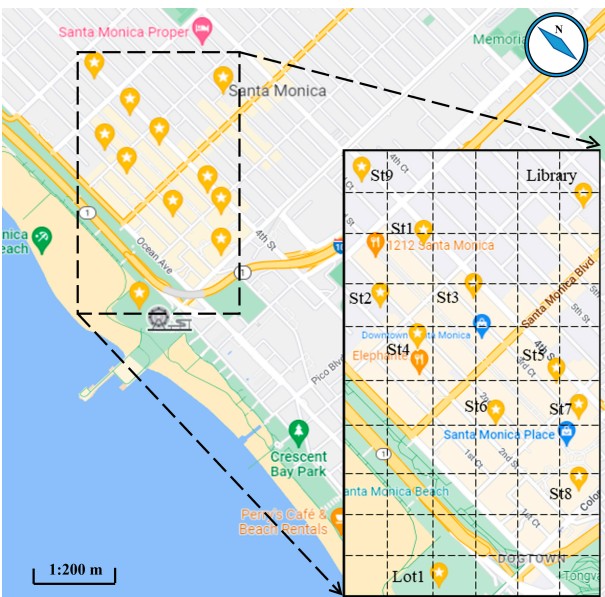

**Figure 1.** Distribution of parking lots in the region.

We use a 100 m × 100 m grid to divide the target area into H × W grids (Figure 1). Each parking lot in the region is distributed in a different grid, and a grid without a parking lot distribution is considered to have no VPSs in that grid. Then, the number of VPSs in the area at time t is denoted as:

$$V_t = \begin{bmatrix} v_t^{(1,1)} & v_t^{(1,2)} & \cdots & v_t^{(1,W)} \\ v_t^{(2,1)} & v_t^{(2,2)} & \cdots & v_t^{(2,W)} \\ \vdots & \vdots & \ddots & \vdots \\ v_t^{(H,1)} & v_t^{(H,2)} & \cdots & v_t^{(H,W)} \end{bmatrix} \tag{1}$$

where each element in the matrix, denoted as $v_t^{(h,w)}$, $h \in [0, H]$, and $w \in [0, W]$, is the number of VPSs in the grid $(h, w)$. This area is divided into a total of 60 grids, with H being 10 and W being 6.

The 11 parking lots we selected in the grid area are the St1–St9 parking lots, Lot1 parking lot, and Library parking lot. These parking lots are mainly distributed in recreational, commercial, and residential areas. It is worth noting that there are similarities and differences in the evolution of the number of spaces in these parking lots. We can imagine that the closer the parking lot types and the closer the distance, the more they should have the characteristics of time–space correlation. We take the Lot1, St5, and St7 parking lots as examples to mine the characteristics of different parking lots from the perspective of spatiotemporal correlation, considering that the St5 and St7 parking lots represent commercial areas and are close to each other, and the Lot1 parking lot represents entertainment areas.

Figures 2 and 3 show the spatiotemporal characteristics of these 3 parking lots. The x-axis represents the time interval. The y-axis represents the change in VPSs, where a positive number represents the outflow of vehicles. The larger the number is, the greater the number of VPSs. Figure 2 shows that the inflow on weekends is significantly higher than that on weekdays during the almost full day in the Lot1 parking lot. The Lot1 parking lot represents an entertainment area, which tends to be crowded on weekends. In contrast, between 7 a.m. and 9 a.m., the inflow of the St5 and St7 parking lots is higher on weekdays than on weekends, which may be influenced by the parking of mall workers. In addition, we explore the impact of weather factors on the availability of VPSs.

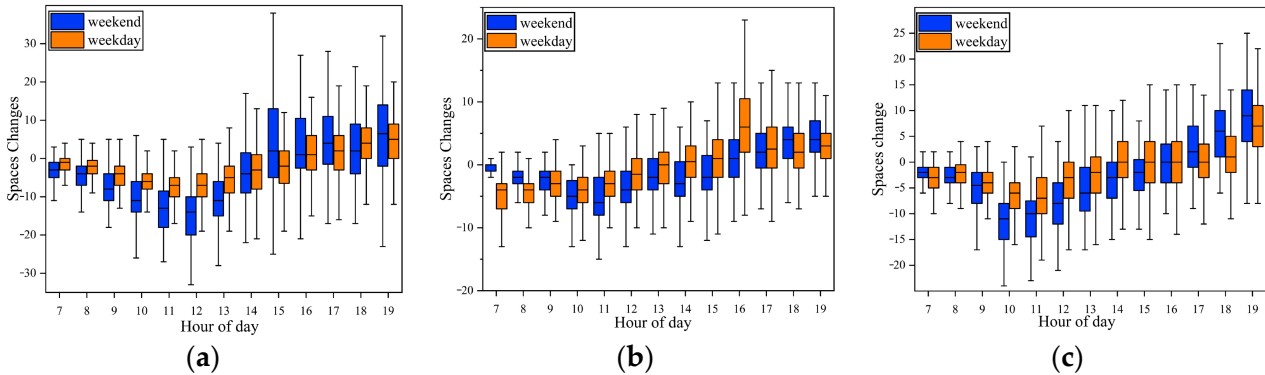

**Figure 2.** Impact of holiday events (weekdays or weekends) on VPS changes (7 a.m. to 7 p.m.). (**a**) Lot1 parking lot; (**b**) St5 parking lot; (**c**) St7 parking lot.

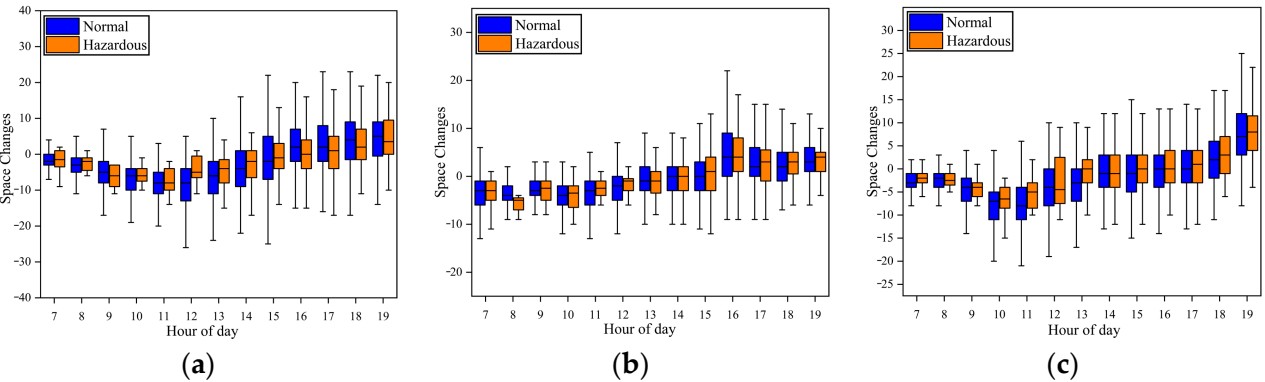

**Figure 3.** Impact of weather on VPS changes (7 a.m. to 7 p.m.). (**a**) Lot1 parking lot; (**b**) St5 parking lot; (**c**) St7 parking lot.

The number of parking occupancies is different for hazardous weather and normal weather. It can be imagined that when encountering hazardous weather such as heavy rain, heavy snow, and smog, people may reduce travel in private cars, so the number of available parking spaces will increase. The results for the three representative parking lots are shown in Figure 3. For the entertainment area represented by Lot1 and the commercial areas represented by St5 and St7, a significant decrease in parking occupancy was observed for all hours of the day under hazardous weather conditions. To assess the impact of hazardous weather conditions on parking demand, we define weather to be considered hazardous if one or more of the following conditions are met: (1) fog or snow, (2) wind speed greater than 39 km/h, (3) precipitation intensity greater than 0.15 inches per hour. All other conditions are considered normal weather conditions. Similar to the research of Yang et al. [38] and Zhao et al. [39], we conduct ablation experiments in Section 4 to explore the influence of external factors such as weather on parking prediction.

### 3.2. Prediction Model

The DWT-ConvGRU-BRC model provided in this study consists of four components, namely, the DWT component, three ConvGRU components, the meta-info feature extraction component, and the BRC component (Figure 4). The first component is the DWT module, which performs noise reduction on VPS data by means of the db3 wavelet basis function. The second component is the ConvGRU module. A CNN can capture spatial correlation well but not temporal correlation. A GRU and LSTM can both model temporal correlation well, but a GRU maintains the prediction accuracy and reduces the running speed compared to LSTM. Therefore, an integration of a CNN and GRU to form a three-layer ConvGRU network can capture both temporal and spatial correlations. The third component is a two-layer linear layer module that incorporates external factors such as temperature, wind

speed, humidity, weekdays, and vacations into the model to enhance the accuracy of long-term forecasts. Finally, feature fusion is performed using the BRC layer to obtain predictions via the sigmoid function.

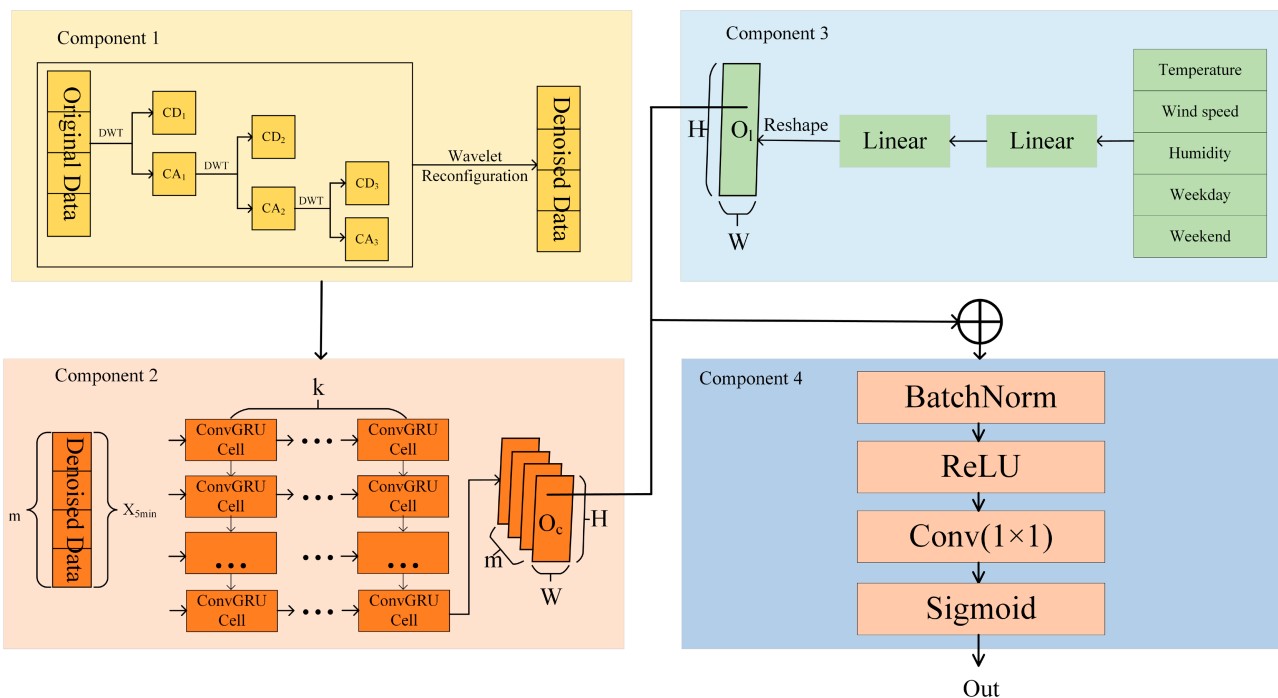

**Figure 4.** DWT-ConvGRU-BRC model.

### 3.2.1. Discrete Wavelet Transform (DWT) Denoising

Time series data obtained in practical applications are often contaminated by various forms and degrees of noise. The discrete wavelet transform is very appropriate for noise filtering, which makes it a good choice for time series data processing [40]. When time series data are decomposed by a DWT, the original signal is separated into approximate coefficients and detail coefficients at different resolution levels. The information of the original signal is retained in the wavelet coefficients, and a perfect reconstruction of the original data can be performed from these coefficients. However, some of the detail coefficients that represent the detailed motion in the data can be identified as noise. These coefficients can then be set to zero prior to the DWT reconstruction process to filter out the noise from the original time series, and reconstruction involves reconstructing the time series from every component except the noise. In other words, a DWT is a discretization of the scales and translations of the fundamental wavelet. A DWT can be defined as:

$$W_\varphi f(m, n) = 2^{-\frac{m}{2}} \int_R f(t)\overline{\psi}(2^{-m}t - n)dt \tag{2}$$

where $\overline{\psi}$ is the complex conjugate of $\psi$, formula $\psi$ satisfies $\int_{-\infty}^{+\infty} \psi(t)dt = 0$, and m and n are integers.

Appropriate wavelet basis functions are very important to extract the features of parking data. Kaplun et al. [41] selected the appropriate wavelet basis function based on entropy estimation in the matching pursuit algorithm. Bhavsar et al. [42] chose the appropriate wavelet basis function by calculating the magnitude of the mutual information. In this paper, we measure the dependence between two variables by calculating the normalized

mutual information (NMI), which places the mutual information between [0, 1], and it is easy to choose a suitable wavelet basis function. The NMI is defined as:

$$NMI(X, Y) = \frac{2(H(X) - H(X|Y))}{H(X) + H(Y)} \tag{3}$$

where $H(X)$ and $H(Y)$ represent the entropy of variables $X$ and $Y$, respectively. $H(X|Y)$ is the conditional entropy for $X$ given $Y$.

In this study, the basisfunctions of the compared wavelets are Daubechies (db3), symlet (sym3), and coiflet (coif3). It can be seen from Figure 5 that db3 has the highest NMI relative to other wavelet functions. After experimental comparison, the db3 wavelet basis is selected for wavelet decomposition of the experimental time series, and the number of decomposition levels is 3, which can remove the noise while maintaining the fluctuation characteristics of the time series data as much as possible.

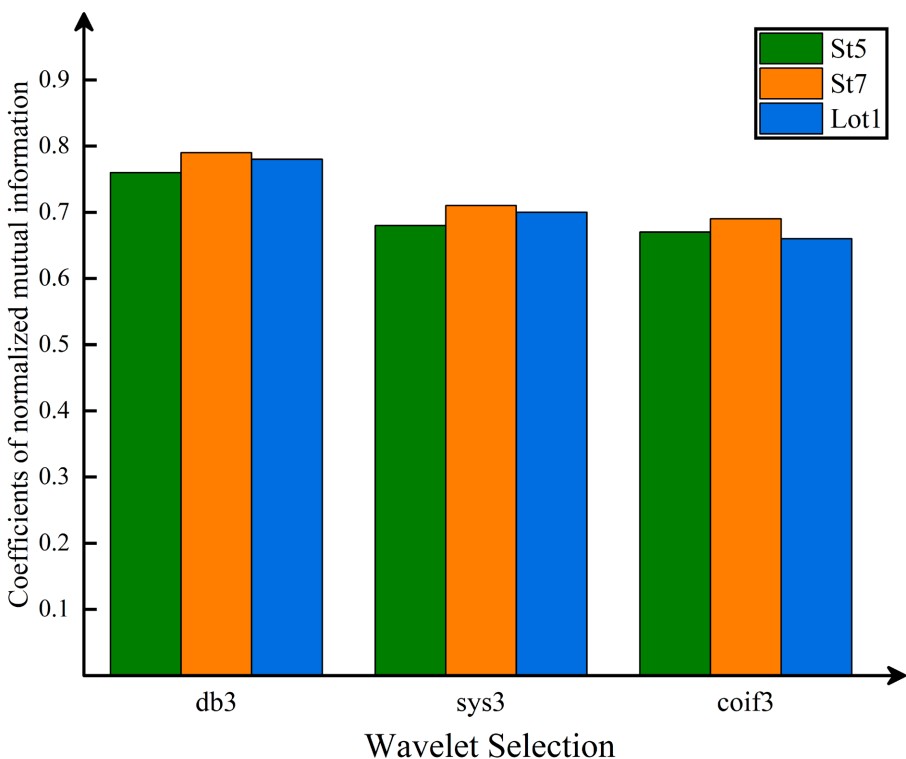

**Figure 5.** Wavelet selection based on normalized mutual information.

We take the St7 parking lot as an example to show the specific process of the DWT component. The original VPS data and its decomposition components are shown in Figure 6. Red is the high-frequency sequence decomposed three times, and blue is the low-frequency sequence. The denoising was performed using the threshold method, and then the denoised time series was reconstructed. Figure 7 shows the comparison between the original and denoised data. We can see that the overall regularity of the denoised data has increased, and the trend tends to be smoother, which is more suitable for model construction. As in Equation (1), the denoised data are $X_t$.

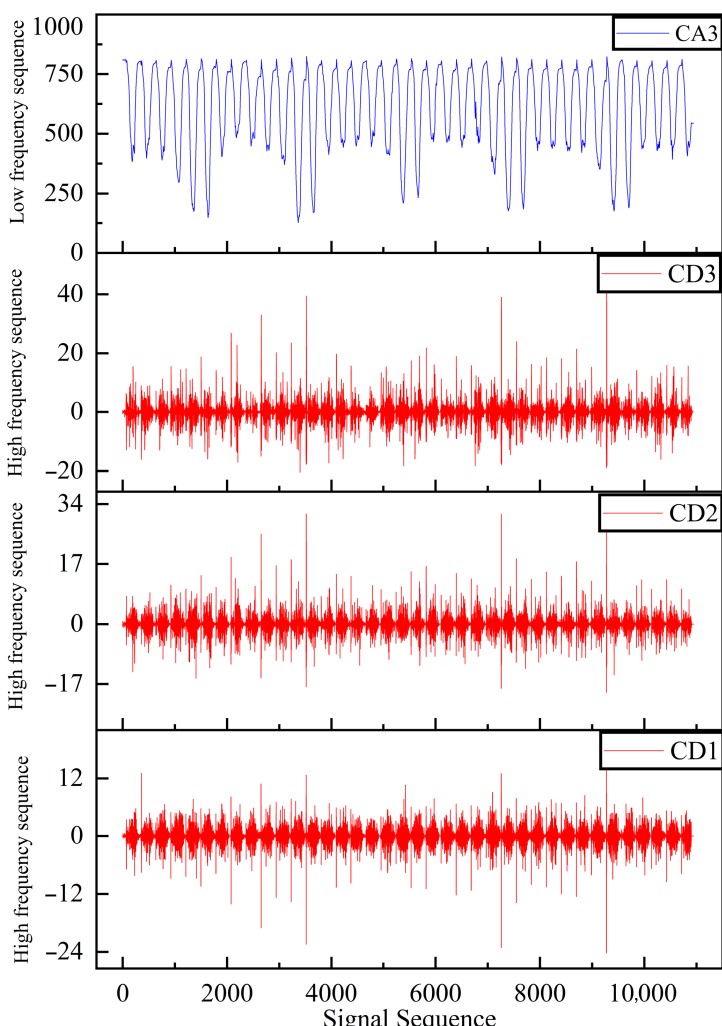

**Figure 6.** db3 wavelet decomposition diagram.

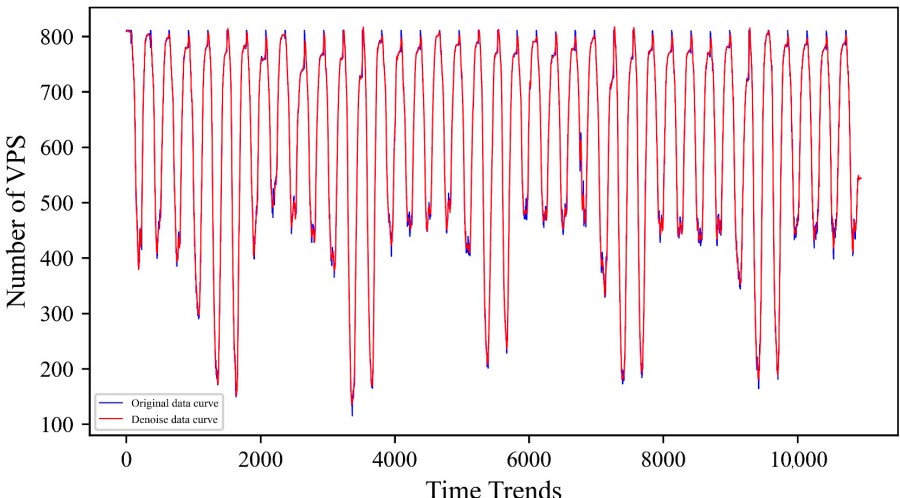

**Figure 7.** Original and denoised time series.

### 3.2.2. Convolutional Gated Recurrent Unit (ConvGRU)

A key difference between a ConvGRU and GRU is that the former uses a convolution operator rather than a fully concatenated operator. Therefore, a ConvGRU can better capture spatial–temporal correlations. Figure 8 shows the internal structure of the ConvGRU

cell, where CAT and UCAT denote the concatenation and splitting operations, respectively. The detailed information flow of a ConvGRU is shown in the following equations:

$$z_t = \sigma(W_{xz} * x_t + W_{hz} * h_{t-1}) \tag{4}$$

$$r_t = \sigma(W_{xr} * x_t + W_{hr} * h_{t-1}) \tag{5}$$

$$\overset{\wedge}{h_t} = \tanh(W_{xh} * x_t + r_t \odot (W_{hh} * h_{t-1})) \tag{6}$$

$$h_t = (1 - z_t) \odot \overset{\wedge}{h_t} + z_t \odot h_{t-1} \tag{7}$$

where $z_t$ denotes the update gate, $r_t$ denotes the reset gate, and $\overset{\wedge}{h_t}$ denotes the candidate hidden state. $x_t$ denotes the input of the current step and $h_t$ and $h_{t-1}$ denote the hidden state of the current and previous steps, respectively. $x_t$ and $h_{t-1}$ are the input and output vectors of the current time point, respectively, and $W_{\cdot z}$, $W_{\cdot r}$, and $W_{\cdot h}$ are the convolutional kernels for each gate. In addition, $\sigma$ and tanh are the sigmoid and hyperbolic tangent activation functions, respectively, $*$ is the convolution operation, and $\odot$ denotes the Hadamard product.

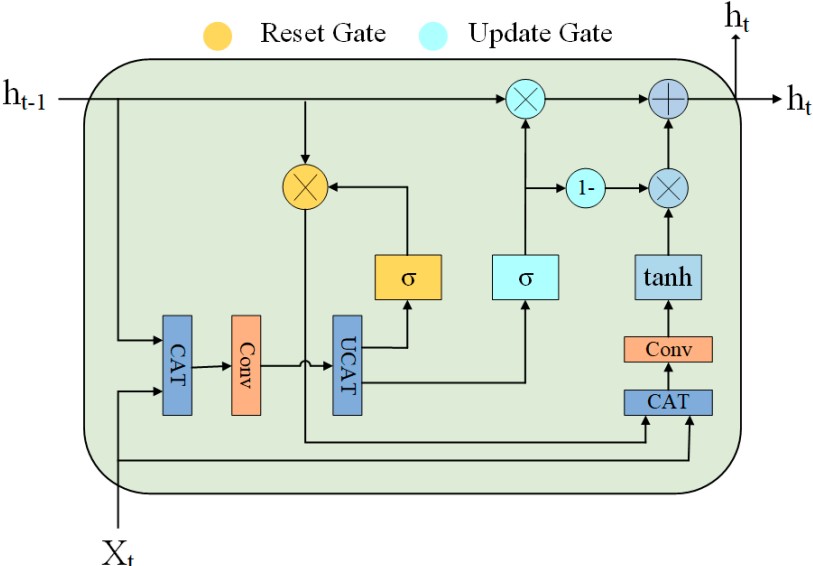

**Figure 8.** Internal structure of the ConvGRU cell.

Specifically, given $X_t$ (the denoised data) and $h_{t-1}$ as inputs at time step t, the unit first obtains the output of the update gate and reset gates, respectively, using Equations (4) and (5). Then, using Equation (6), the temporal hidden state $\overset{\wedge}{h_t}$ can be calculated, which considers both the input $X_t$ and the hidden state $h_{t-1}$ produced by the previous ConvGRU operator. The final hidden state $h_t$ of the unit is produced by a linear combination of temporal hidden state $\overset{\wedge}{h_t}$ and previous hidden state $h_{t-1}$ using Equation (7). We denote the output of this part as follows:

$$O_c = h_t \oplus \cdots \oplus h_{t-5(m-2)} \oplus h_{t-5(m-1)} \tag{8}$$

where *m* denotes the number of data divided into 5 min intervals and $\oplus$ is the concatenation operation.

### 3.2.3. External Factor Extraction and Feature Learning

Parking spaces have obvious periodic characteristics. We analysed the impact of external factors, such as weekdays, non-weekdays, hazardous weather, and normal weather,

on the availability of parking spaces. External factors can affect parking events and are an important part of the model. A two-layer linear layer was designed to consider the impact of external factors on VPSs. We record the data collected from [43] on temperature, wind speed, humidity, etc., and weekdays and non-weekdays as inputs to this section as n. Bringing n into Equation (9), we obtain $O_l \in \mathbb{R}^{HW \times 1}$:

$$O_l = \sigma(w_{n,2}\sigma(w_{n,1}n + b_{n,1}) + b_{n,2}) \tag{9}$$

where $\sigma$ is the activation function and $w_{n,i}, b_{n,i}, i = 1, 2$ are the weights and deviations of the $i$-th linear function. For feature learning, we convert the resulting output $O_l \in \mathbb{R}^{HW \times 1}$ to $O_l \in \mathbb{R}^{H \times W}$ via the Reshape function.

The outputs of the three ConvGRUs and the additional factor extraction component are concatenated together, denoted as $O_i = O_c \oplus O_l$, and fed into the BRC layer. The BRC layer is a composite function of three consecutive operations, i.e., batch normalization (BN), rectified linear activation (ReLU), and a $1 \times 1$ convolution (Conv). We use the BRC layer to implement feature reuse and propagation. Finally, the prediction is obtained by applying the sigmoid function.

## 4. Experimental Results

### 4.1. Experimental Setup and Evaluation Indicators

In our experiments, we select 60% of the data as the training set, 20% as the validation set, and the rest as the test set. We normalize the denoised data using Equation (10) and then slice it into the model using single-step moving window data of length 10. For training, the gradient descent optimization algorithm is the Adam [44] algorithm, the learning rate is 0.01, the loss function is the MAE, the epoch size is 32, the batch size is 32, and the number of ConvGRU layers (k) is 3. The numbers of VPSs after 5, 15, 30, 45, and 60 min are predicted accordingly. To avoid contingency, each prediction task was independently repeated 30 times, and the mean values were taken as the results. The DWT-ConvGRU-BRC model is implemented using PyTorch version 1.11.0, and the experimental equipment includes a 12th Gen lntel(R) Core(TM) i5-12600KF processor and an NVIDIA GeForce GTX 30600Ti GPU with 16 GB memory.

$$x^* = \frac{x - \min}{\max - \min} \tag{10}$$

where max and min represent the maximum and minimum values of the sample data, respectively, and max–min represents the range.

We use the mean absolute error (MAE), mean absolute percentage error (MAPE), and root mean square error (RMSE) to measure the accuracy of the predicted values. All three evaluation metrics have a range of [0, +∞) and are equal to 0 when the predicted value exactly matches the true value, i.e., a perfect model; the larger the error is, the larger the value. The definitions are shown in Equations (11)–(13).

$$MAE = \frac{1}{n}\sum_{i=1}^{n}|\hat{y}_i - y_i| \tag{11}$$

$$MAPE = \frac{100\%}{n}\sum_{i=1}^{n}\frac{|\hat{y}_i - y_i|}{y_i} \tag{12}$$

$$RMSE = \sqrt{\frac{1}{n}\sum_{i=1}^{n}(\hat{y}_i - y_i)^2} \tag{13}$$

where $y_i$ denotes the actual VPSs, $\hat{y}_i$ denotes the predicted VPSs, and $n$ denotes the time step.

## 4.2. Results and Analysis

Fifty experiments were carried out, and the mean was selected as the result to improve the statistical significance of the difference in precision. Table 1 shows the detailed RMSE, MAE, and MAPE data for our DWT-ConvGRU-BRC model. Figure 9 shows the effect of the DWT-ConvGRU-BRC model in the Lot1, St5, and St7 parking lots for 5, 15, 30, 45, and 60 min VPS predictions, where the x-axis represents the time interval, and the y-axis represents the number of VPSs. Our model is robust in predicting the availability of VPSs in multiple parking lots.

**Table 1.** Performance evaluation in terms of the MAE, RMSE, and MAPE.

| | | DWT-ConvGRU-BRC without External Factors | | | | | DWT-ConvGRU-BRC with External Factors | | | | |
|---|---|---|---|---|---|---|---|---|---|---|---|
| | | 5 min | 15 min | 30 min | 45 min | 60 min | 5 min | 15 min | 30 min | 45 min | 60 min |
| RMSE | St1 | 2.87 | 4.59 | 5.55 | 8.10 | 9.48 | 2.93 | 3.76 | 5.81 | 7.21 | 9.03 |
| | St2 | 5.72 | 4.56 | 7.26 | 8.88 | 10.98 | 3.22 | 4.31 | 6.70 | 9.29 | 10.26 |
| | St3 | 3.31 | 4.14 | 6.13 | 8.42 | 10.78 | 3.21 | 3.80 | 6.20 | 8.06 | 9.44 |
| | St4 | 3.74 | 4.97 | 8.52 | 9.75 | 13.90 | 3.65 | 4.74 | 7.46 | 10.43 | 11.80 |
| | St5 | 4.16 | 4.97 | 8.43 | 12.57 | 14.38 | 3.77 | 5.60 | 8.92 | 12.08 | 14.64 |
| | St6 | 5.20 | 6.17 | 9.47 | 12.12 | 15.00 | 4.86 | 6.07 | 9.79 | 11.98 | 14.83 |
| | St7 | 5.20 | 6.99 | 12.14 | 16.90 | 21.40 | 5.07 | 6.49 | 10.44 | 15.33 | 19.35 |
| | St8 | 6.14 | 7.21 | 12.18 | 16.60 | 22.18 | 5.42 | 6.97 | 11.14 | 16.38 | 20.74 |
| | St9 | 2.63 | 3.32 | 4.03 | 4.67 | 5.40 | 1.98 | 3.09 | 4.06 | 4.87 | 5.49 |
| | Lot1 | 9.21 | 12.27 | 18.84 | 34.60 | 41.72 | 9.19 | 12.01 | 19.54 | 27.38 | 34.50 |
| | Library | 2.48 | 2.28 | 3.34 | 3.65 | 4.49 | 2.32 | 2.28 | 3.73 | 4.58 | 5.23 |
| MAE | St1 | 2.01 | 2.91 | 3.85 | 5.64 | 6.48 | 2.08 | 2.58 | 4.00 | 5.13 | 6.19 |
| | St2 | 3.22 | 4.31 | 6.70 | 9.29 | 10.26 | 2.29 | 3.12 | 4.91 | 6.51 | 7.44 |
| | St3 | 2.28 | 2.89 | 4.34 | 5.88 | 7.27 | 2.25 | 2.65 | 4.44 | 5.59 | 6.76 |
| | St4 | 2.67 | 3.54 | 5.88 | 6.77 | 9.89 | 2.56 | 3.40 | 5.33 | 7.24 | 8.17 |
| | St5 | 3.11 | 3.51 | 5.85 | 8.71 | 10.01 | 2.62 | 4.01 | 6.11 | 8.03 | 10.20 |
| | St6 | 3.77 | 4.36 | 6.88 | 8.77 | 11.43 | 3.43 | 4.36 | 6.99 | 8.80 | 11.08 |
| | St7 | 3.66 | 4.99 | 8.42 | 11.54 | 14.03 | 3.60 | 4.65 | 7.41 | 10.58 | 12.89 |
| | St8 | 4.09 | 4.93 | 8.35 | 11.10 | 14.26 | 3.80 | 4.97 | 7.79 | 11.38 | 13.78 |
| | St9 | 1.81 | 2.38 | 2.86 | 3.32 | 3.83 | 1.42 | 2.30 | 2.86 | 3.47 | 4.06 |
| | Lot1 | 5.57 | 7.77 | 11.75 | 21.60 | 25.53 | 5.67 | 7.68 | 12.67 | 18.22 | 23.18 |
| | Library | 1.59 | 1.43 | 2.18 | 2.40 | 3.04 | 1.64 | 1.56 | 2.30 | 3.00 | 3.19 |
| MAPE (%) | St1 | 0.82 | 1.24 | 1.56 | 2.30 | 2.64 | 0.84 | 1.07 | 1.65 | 2.07 | 2.51 |
| | St2 | 0.76 | 0.63 | 1.01 | 1.25 | 1.52 | 0.44 | 0.60 | 0.94 | 1.25 | 1.43 |
| | St3 | 1.04 | 1.37 | 2.03 | 2.75 | 3.37 | 1.01 | 1.20 | 2.00 | 2.55 | 3.08 |
| | St4 | 0.56 | 0.74 | 1.26 | 1.46 | 2.12 | 0.54 | 0.71 | 1.13 | 1.56 | 1.76 |
| | St5 | 0.66 | 0.78 | 1.30 | 1.90 | 2.20 | 0.58 | 0.88 | 1.34 | 1.79 | 2.21 |
| | St6 | 0.70 | 0.84 | 1.33 | 1.68 | 2.18 | 0.64 | 0.82 | 1.32 | 1.66 | 2.11 |
| | St7 | 0.79 | 1.10 | 1.83 | 2.57 | 2.96 | 0.78 | 0.97 | 1.56 | 2.26 | 2.75 |
| | St8 | 0.62 | 0.75 | 1.27 | 1.69 | 2.19 | 0.57 | 0.74 | 1.17 | 1.67 | 2.08 |
| | St9 | 0.74 | 0.96 | 1.16 | 1.34 | 1.55 | 0.57 | 0.92 | 1.16 | 1.40 | 1.63 |
| | Lot1 | 0.68 | 0.93 | 1.60 | 3.49 | 4.14 | 0.69 | 0.95 | 1.64 | 2.32 | 3.07 |
| | Library | 0.32 | 0.28 | 0.44 | 0.48 | 0.60 | 0.33 | 0.31 | 0.46 | 0.60 | 0.64 |

From Section 3, we conclude that parking occupancy is affected by external factors such as weather. This is consistent with the conclusions drawn in [33–35,38,39]. Therefore, we performed ablation experiments in Table 1 to compare the effect of the presence or absence of external factors on VPS prediction. We find that the inclusion of external factors such as temperature, wind speed, weekdays, and weekends was beneficial in improving the accuracy of the forecasts. For the RMSE, MAE, and MAPE, the forecasting models with external factors outperformed those without external factors in 40, 35, and 39 of the 55 forecasting tasks, respectively. This shows that considering external factors can improve the accuracy of VPS prediction, so it can be said that external factors have a significant

impact on parking prediction. Figure 10 specifically shows the prediction effect of our proposed DWT-ConvGRU-BRC model for each parking lot. It can be seen from the figure that our model has good timeliness in predicting the number of VPSs in multiple parking lots, and it is relatively robust to changes in the spatial–temporal correlation of VPSs and can accurately achieve predictions.

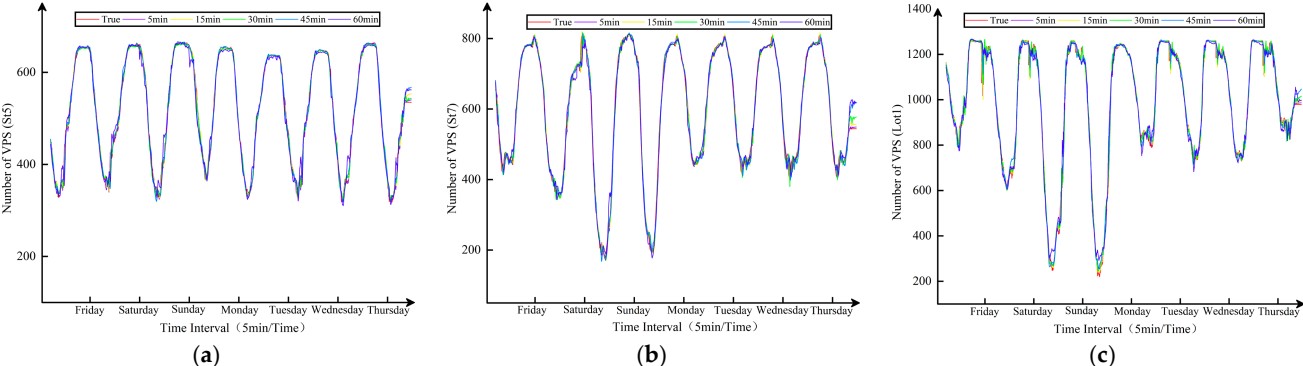

**Figure 9.** Comparison between the DWT-ConvGRU-BRC prediction and actual values at 5, 15, 30, 45, and 60 min. (**a**) St5 parking lot; (**b**) St7 parking lot; (**c**) Lot1 parking lot.

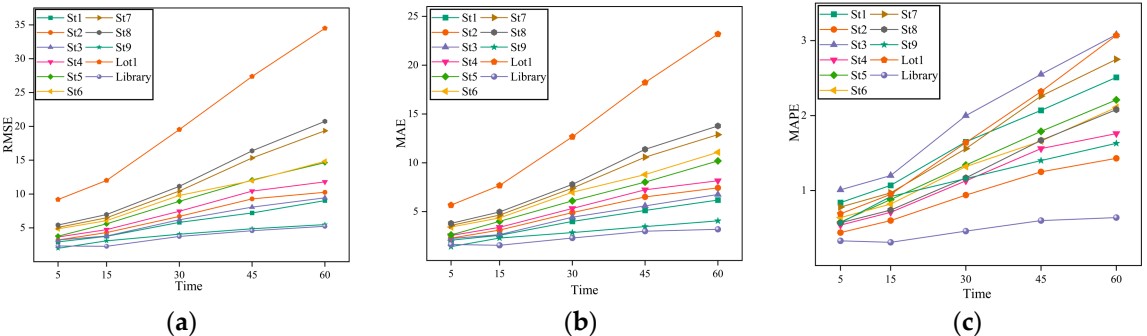

**Figure 10.** DWT-ConvGRU-BRC model evaluation metrics for each parking lot. (**a**) RMSE; (**b**) MAE; (**c**) MAPE.

As some models can only predict for a single parking lot, we compared the effectiveness of these models in predicting the number of VPSs at 5, 15, 30, 45, and 60 min, using the St7 parking lot as an example. The LSTM and GRU models can effectively extract temporal information from nonlinear time series data, but they fail to consider the spatial correlation between parking lots within a region. The ConvGRU model outperforms the ConvLSTM model in terms of running speed while capturing spatial–temporal correlations. After wavelet noise reduction, the forecasts improved significantly, and external factors, such as weather, improved the accuracy of the long-term forecasts. As illustrated in Table 2, the proposed DWT-ConvGRU-BRC model is significantly superior to the benchmark methods.

**Table 2.** Comparison of operating results.

| Model | Indicator | 5 min | 15 min | 30 min | 45 min | 60 min |
|---|---|---|---|---|---|---|
| LSTM | RMSE | 14.86 | 17.14 | 20.84 | 29.24 | 34.45 |
| | MAE | 8.54 | 11.07 | 14.92 | 26.71 | 33.12 |
| | MAPE(%) | 1.32 | 1.72 | 2.31 | 4.15 | 5.17 |
| GRU | RMSE | 10.41 | 13.86 | 18.75 | 23.42 | 29.54 |
| | MAE | 7.80 | 11.40 | 15.15 | 19.25 | 21.74 |
| | MAPE(%) | 1.57 | 2.50 | 2.93 | 4.36 | 5.07 |

**Table 2.** *Cont.*

| Model | Indicator | 5 min | 15 min | 30 min | 45 min | 60 min |
|---|---|---|---|---|---|---|
| | RMSE | 7.14 | 10.46 | 14.41 | 17.83 | 21.73 |
| ConvGRU | MAE | 4.74 | 7.25 | 10.01 | 12.31 | 14.59 |
| | MAPE(%) | 1.01 | 1.51 | 2.08 | 2.62 | 3.16 |
| | RMSE | 7.70 | 10.55 | 14.28 | 16.87 | 20.01 |
| ConvLSTM | MAE | 5.15 | 7.30 | 9.91 | 11.45 | 13.14 |
| | MAPE(%) | 1.09 | 1.54 | 2.07 | 2.35 | 2.73 |
| | RMSE | 7.23 | 10.75 | 15.10 | 17.88 | 19.43 |
| dConvLSTM-DCN | MAE | 4.98 | 7.36 | 10.18 | 11.77 | 13.02 |
| | MAPE(%) | 1.04 | 1.53 | 2.11 | 2.47 | 2.68 |
| | RMSE | 5.2 | 6.99 | 12.14 | 16.9 | 21.4 |
| DWT-ConvGRU | MAE | 3.66 | 4.99 | 8.42 | 11.54 | 14.03 |
| | MAPE(%) | 0.79 | 1.1 | 1.83 | 2.57 | 2.96 |
| | RMSE | 5.07 | 6.49 | 10.44 | 15.33 | 19.35 |
| DWT-ConvGRU-BRC | MAE | 3.6 | 4.65 | 7.41 | 10.58 | 12.89 |
| | MAPE(%) | 0.78 | 0.97 | 1.56 | 2.26 | 2.75 |

To illustrate the ability of the proposed model to predict the actual number of VPSs, in Table 3, we present a detailed comparison between the actual and predicted number of St7 parking lot VPSs (from 10:00 to 11:00, 6 May 2021) output by the proposed DWT-ConvGRU-BRC model. We also calculate the MAE, MAPE, and RMSE values for the time period. The output values of the DWT-ConvGRU-BRC model are very close to the real values.

**Table 3.** Comparisons between the real and predicted numbers of VPSs at the St7 parking lot.

| Time | Real | Predicted Values | | | | |
|---|---|---|---|---|---|---|
| Point | Value | 5 min | 15 min | 30 min | 45 min | 60 min |
| 6 May 2021 10:00 | 652 | 647 | 643 | 654 | 674 | 669 |
| 6 May 2021 10:05 | 652 | 645 | 639 | 643 | 667 | 661 |
| 6 May 2021 10:10 | 644 | 646 | 636 | 632 | 659 | 655 |
| 6 May 2021 10:15 | 641 | 642 | 633 | 620 | 648 | 645 |
| 6 May 2021 10:20 | 643 | 641 | 632 | 622 | 637 | 636 |
| 6 May 2021 10:25 | 639 | 641 | 629 | 617 | 625 | 626 |
| 6 May 2021 10:30 | 638 | 642 | 625 | 614 | 611 | 615 |
| 6 May 2021 10:35 | 628 | 632 | 624 | 615 | 597 | 604 |
| 6 May 2021 10:40 | 615 | 623 | 624 | 606 | 593 | 591 |
| 6 May 2021 10:45 | 609 | 613 | 616 | 605 | 587 | 578 |
| 6 May 2021 10:50 | 603 | 600 | 606 | 605 | 582 | 564 |
| 6 May 2021 10:55 | 594 | 594 | 596 | 605 | 582 | 557 |
| 6 May 2021 11:00 | 584 | 584 | 585 | 591 | 575 | 554 |
| MAE | | 3.23 | 7.54 | 12.08 | 17.15 | 20.69 |
| MAPE(%) | | 0.51 | 1.19 | 1.91 | 2.74 | 3.36 |
| RMSE | | 4.01 | 8.45 | 14.17 | 18.69 | 23.46 |

In addition, model performance evaluation should consider both prediction accuracy and time consumption. Table 4 compares the running times per round of the LSTM, GRU, ConvGRU, ConvLSTM, dConvLSTM-DN, DWT-ConvGRU, and DWT-ConvGRU-BRC models. The DWT-ConvGRU-BRC model considers the effects of factors such as wavelet noise reduction and external factors, so it is slightly inferior to the models that do not consider these factors in terms of running speed, but compared to the dConvLSTM-DN model proposed in [36], our model shows a significant improvement in running speed. In conclusion, our proposed model not only improves the effectiveness of predictions but also improves the running speed.

**Table 4.** GPU runtimes of different prediction methods.

| Model | LSTM | GRU | ConvLSTM | ConvGRU |
| --- | --- | --- | --- | --- |
| Runtime (s/epoch) | 3.34 | 2.46 | 17.21 | 15.07 |
| Model | dConvLSTM-DCN | ConvGRU-BRC | DWT-ConvGRU | DWT-ConvGRU-BRC |
| Runtime (s/epoch) | 21.82 | 15.51 | 16.04 | 16.28 |

## 5. Conclusions

This paper proposes a deep learning model for occupancy prediction of multiple parking lots. The model incorporates DWT, ConvGRU, and BRC modules and has the flexibility to take multiple spatial–temporal structured data sources as inputs. The performance of the model is evaluated using a case study from 11 public parking lots in Santa Monica, California, USA, in which VPS data, weather data, and weekday and weekend data are used. The experimental results show that our model can achieve considerably high accuracy with MAPEs of less than 2% for short-term predictions and less than 4% for long-term predictions. The DWT-ConvGRU-BRC model significantly outperforms the baseline LSTM and GRU methods. In general, we found that noise reduction of VPS data using a DWT can improve prediction accuracy and that combining weather information and weekday and weekend information can improve the performance of long-term predictions of parking occupancy.

Prediction of available parking spaces is an integral part of parking guidance information systems. Available parking space predictions can improve the effectiveness of parking guidance system information, which can help drivers plan driving routes and find vacant parking spaces. Furthermore, if we have a reliable parking prediction algorithm, we can apply dynamic parking pricing to control the parking demand of each parking lot, thereby assisting traffic planning and management and reducing energy consumption and traffic congestion. In future work, we will concentrate on further improving the adaptability by considering other external influences, such as POI information, traffic incident data, traffic flow data, etc. At the same time, future research will also consider how the running time of the model can be optimized while ensuring prediction accuracy.

**Author Contributions:** Methodology, W.J.; Formal analysis, Z.H.; Resources, W.J.; Writing—original draft, W.F.; Writing—review & editing, L.G. All authors have read and agreed to the published version of the manuscript.

**Funding:** This research was funded by Fujian Natural Science Foundation (Project No.: 2020J05194, 2021J05226, and 2022J01938).

**Institutional Review Board Statement:** Not applicable.

**Informed Consent Statement:** Not applicable.

**Data Availability Statement:** Not applicable.

**Conflicts of Interest:** The authors declare no conflict of interest.

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
