# Peer review of "Prediction of Vacant Parking Spaces in Multiple Parking Lots: A DWT-ConvGRU-BRC Model"

_applsci, doi:10.3390/app13063791_

Round 1

Reviewer 1 Report

The study describes the deep learning model called DWT-ConvGRU-BRC to predict the 12 future availability of VPSs in multiple parking lots. The study is unique in its nature and has shown a good approach to predicting the availability of parking lots using deep learning models. But I have the following suggestions to further improve its readability to the readers.

1. The first line of the abstract should be revised in a more simple form.

2. the abstract should be divided into subsections. like background, methodology, results, and conclusion.

3. introduction section should be further extended and literature review should be separated from it.

4. MAE, RMSE, and MAPE are extensively used for model comparison purposes. Is there any other method that can also be added for comparison purposes like scatter index, bias, and DR?

5. Conclusion section should be rewritten with results, discussion, and future implications, 

6. Is there any limitation of the study means if  weather information and weekday and  weekend information is different in other situation will such people able to outperformed other models.

Author Response

On behalf on my co-authors and myself, I sincerely thank you and the 
anonymous reviewers of our manuscript submitted under the title “Prediction of Vacant Parking Spaces in Multiple Parking Lots: A DWT-ConvGRU-BRC 
Model” for a thoughtful review and many insightful comments. We have made 
substantial revision to the paper, and are now pleased to resubmit it in what we 
believe is an improved paper.
Because the edits and changes made to the original manuscript were 
numerous and extensive, we did not use track changes to highlight all revisions. 
Instead, we choose to highlight the major revisions using blue color texts. 
Specifically, the major improvements in the revised version include:
• Revised the title to “Prediction of Vacant Parking Spaces in Multiple 
Parking Lots: A DWT-ConvGRU-BRC Model”. Rewrote the Introduction 
section and Literature Review section.
• Measured the dependence between two variables by calculating the 
normalized mutual information (NMI), and chose the appropriate wavelet basis 
function by comparing the size of the NMI. The comparison results were shown 
in Figure3, so we chose db3 wavelet as the basis function of discrete wavelet 
transform.
• Appropriate explanations had been added to Figure 3 in the manuscript to 
facilitate understanding. In the revised version, we made changes to Figure 8. 
We put the true and 5 predicted values in one plot to show the multi-step 
forecasting performance of our model.
• Added discussion part and future implications in the results section to 
improve the structural organization of this paper.
Meanwhile, we have also carefully re-edited our wordings throughout the 
paper to sharpen the article expression. Specific point-to-point response to 
review comments is attached below. Thank you again for your attention and 
consideration. We look forward to hearing from you. Please see the attachment

Reviewer 2 Report

Authors proposed A DWT-ConvGRU-BRC Model to predict Vacant Parking Space Availability. I suggest certain modifications as well as technical justifications required based on following comments:

1. Title should be modified. Word "Regional Level Perspective" seems to be not fit in title.

2. As compared to published literature, what is author's specific contribution. Kindly add in Introduction section.

3. There are numerous wavelets available and various criteria suggested for wavelet selection. In submitted manuscript,authors used db3 wavelet.What is the specific reason to choose only this wavelet. Suggest to refer following journals for better clarity and add with suitable description in revised version:

a. https://www.mdpi.com/2075-1702/10/3/176

b. https://www.mdpi.com/1099-4300/21/9/843

4. Fig.3 explanation is required for better understanding in revised version.

5. Resolution of Fig.8 need to be improved.Further use same font size and styles for all figures.

6. Influence of external factors as mentioned in Table 1 should be explained in detail.

7. While looking towards Fig.8 all figures seems to be same or in some figure there are least variation. What is the utility of fig.8.Kindly add suitable discussion.

Author Response

(The authors gave the same response as above.)

Round 2

Reviewer 2 Report

Authors have incorporated the suggestions given by reviewer in revised version and justified the comments.